# Behavioral Strengths and Difficulties and Their Associations with Academic Performance in Math among Rural Youth in China

**DOI:** 10.3390/healthcare10091642

**Published:** 2022-08-28

**Authors:** Wenjing Yu, Cody Abbey, Yiwei Qian, Huan Wang, Scott Rozelle, Manpreet K. Singh

**Affiliations:** 1China Academy for Rural Development, Zhejiang University, Hangzhou 310058, China; 2Stanford Center on China’s Economy and Institutions, Stanford University, Stanford, CA 94305-6055, USA; 3Research Institute of Economics and Management, Southwestern University of Finance and Economics, Chengdu 610074, China; 4Stanford School of Medicine, Stanford University, Stanford, CA 94305-5101, USA

**Keywords:** behavioral problem, mental health, academic performance, developing countries, rural China

## Abstract

Behavioral strengths and difficulties among children and adolescents may be significantly associated with their academic performance; however, the evidence on this issue for rural youth in developing contexts is limited. This study explored the prevalence and correlates of mental health from three specific dimensions—internalizing problems, externalizing problems, and prosocial behavior—measured by the Strengths and Difficulties Questionnaire (SDQ), and the association of these dimensions with academic performance in math among a sample of 1500 students in rural China. Our findings indicated that students in rural China had worse behavioral difficulties and poorer prosocial skills when compared to most past studies conducted inside and outside of China. In addition, total difficulties and prosocial scores on the SDQ were significantly associated with student math test scores, as students whose externalizing, internalizing, and prosocial scores were in the abnormal range scored lower in math by 0.35 SD, 0.23 SD, and 0.33 SD, respectively. The results add to the growing body of empirical evidence related to the links between social environment, mental health, and academic performance in developing countries, highlighting the importance of students’ mental health for their academic performance, and of understanding risk factors in the social environment among rural youth in developing countries.

## 1. Introduction

Globally, 10–20% of children and adolescents experience some form of mental illness [1]. Although the majority of school-aged children live in low- and middle-income countries (LMICs) [2,3], the current literature largely focuses on exploring the mental health of students in high-income countries [4,5,6,7,8]. There remains a need for further understanding of the prevalence and consequences of poor mental health in developing countries and regions, in order to better design and target policy, allocate resources, and provide support to mitigate mental illness and optimize mental health among vulnerable student populations.

Low academic achievement is an unfortunate consequence of mental illness, and often leads to treatment-seeking among school-age youth [9,10,11,12]. In contrast, mental health has been positively associated with favorable academic outcomes [13,14]. Indeed, low levels of academic performance have been associated with high levels of mental health problems during elementary school and middle school [15].

When assessing the association between students’ mental health and academic achievement, it is important to consider the multiple dimensions that comprise mental health and illness. Past studies in developed contexts have indicated that both internalizing problems (referring to internal symptoms such as depression, misery, worries, fear, hypochondriasis, and obsessions) [16]) and externalizing problems (referring to external behaviors such as propensity for significant outbursts, lying, stealing, defiance, disobedience, truancy, and property distribution) [16]) are negatively associated with student academic functioning in the early years. In contrast, prosocial ability—defined as proactively seeking and sustaining relationships, which may involve skills such as prosocial moral reasoning, social competence, and self-regulation [17])—has a positive association with academic performance [18,19]. Likewise, several longitudinal studies have also implicated causal relations between these component dimensions of mental health. For instance, externalizing problems evident in childhood appear to undermine student academic performance, which appears to then increase the likelihood of internalizing problems later on in young adulthood [15,20,21]. Additionally, social–emotional interventions designed to improve student prosocial skills have also exhibited positive effects on academic performance in the short term [22], which can also have a strong positive impact on children’s later academic achievement [23].

However, research in developing countries concerning the link between children and adolescents’ mental health problems and their academic performance is still limited. A small handful of studies—such as one in Ghana [24] and another in Iran [25]—in such contexts have demonstrated that child and adolescent mental health problems were associated with significantly poorer academic performance; however, both the Ghana and the Iran studies had a number of limitations. Firstly, they had limited data, in that they had small sample sizes, and did not consider the impact of socio-economic factors that may have moderated the relationship between mental health and academic performance. Secondly, they focused on a specific mental health condition, such as depression, and this may have limited their inferences about other conditions that may have been more prevalent (such as anxiety), or about overall mental health. An even smaller number of studies—such as one in Chile [26] and another in Pakistan [27]—used longitudinal designs to follow the association of mental health problems and academic performance over the course of several years of child development; however, these studies primarily focused on the impact of interventions aimed at improving mental health, and could not fully describe the relationship between mental health and academic performance, due a limited number of social–environmental variables.

In addition to the small number of existing studies, and the limited range of contexts where current research has been conducted in developing countries, another limitation of the current literature on developing countries is that existing studies often do not separately measure the association of different mental health dimensions (i.e., internalizing symptoms, externalizing symptoms, and prosocial behaviors) and academic performance [28]. There is robust evidence that internalizing problems (such as emotional and peer relationship problems) are directly related to students’ academic difficulties [13,29]. The literature has also linked externalizing problems (such as thought or attention problems) and a lack of prosocial skills to academic underachievement [30,31], though less consistently. Thus, more needs to be learned about the interplay between children and specific dimensions of adolescent mental health and academic achievement in low- and middle-income countries (LMICs).

Rural China is an ideal setting to study the above issues. The socio-economic status of China’s rural population lags far behind that of urban residents, and its mental healthcare and educational resources are also of lower quality [32,33]. This lack of resources is accompanied by poorer academic and mental health outcomes among students in rural areas, where approximately 70% of students grow up in China [34,35]. Compared with urban students, children and adolescents in rural areas are more likely to be poor academic performers, especially in arithmetic and mathematics [36,37,38]—the core subjects in school education—which are more explicitly tied to learning experiences at school, as opposed to learning experiences at home (such as reading or language achievement) [39]. In addition, rural students are at greater risk of experiencing externalizing and internalizing problems [40], and the prevalence of deficits in prosocial behaviors is also much higher than among urban students [41]. There are potential targets for interventions to encourage youth in rural China to flourish [42], so understanding the relationship between mental health and academic performance is all the more critical.

It is possible that there is a link between poor mental health and poor academic performance in rural China, especially in math; however, few empirical studies have explored this association while controlling for sociodemographic characteristics. That is not to say that there are no studies on mental health among rural students in China; however, most of those studies have solely examined prevalence and determinants of mental health. Likewise, a wide range of psychosocial factors—including low family income [43], low family support [44], less parental supervision at home [45], less involvement in activities at school [46], boarding at school [47], and peer-to-peer bullying [48]—have all been found to be related to poor academic performance; however, few studies have identified the association between mental health and academic performance in rural China, especially by using a multi-dimensional measure of mental health.

To fill these gaps in the literature, the present study focused on the association between student mental health and academic performance in math in rural China. Our specific objectives were to: (a) measure the mental health (including internalizing problems, externalizing problems, and prosocial behavior) of a large sample of children and adolescents in rural China, using standardized and internationally validated scales; (b) identify the social–environmental factors associated with student mental health; and (c) measure the association between student mental health conditions and academic performance in math, controlling for gender, age, family functioning, peer victimization, and family assets as the possible factors that may moderate the relationship between mental health and academic performance.

## 2. Materials and Methods

### 2.1. Sampling

This study used cross-sectional data collected in October 2020 from 30 rural schools in Gansu province, located in northwest China. Per capita yearly income of households in the sample region was approximately 1354 USD, which was significantly lower than the national average income of rural Chinese residents (2249 USD), and lay in the second-lowest income quintile among the rural population [35]. Moreover, approximately 58% of the population in the sample area were rural residents, which was higher than the share of China’s overall rural population (40%) [49].

Our sample was selected by following a two-step protocol. In the first step, 20 elementary schools and 10 junior high schools were selected randomly from all the rural schools in the county. Based on power calculation, a total of 30 schools were needed to reach 80% statistical power. Due to limited funding, two classes at most were randomly selected from the fourth, fifth or seventh and eighth grades of each sample school, unless there were only one or two classes in a grade. We did not include the sixth grade, as those students were preparing for middle-school entrance examinations, and it would therefore have been difficult to gain approval to conduct a survey amongst them.

The second step was to select sample students. Half of the students in each sample class, who were present on the day of the survey, were randomly selected to participate in the survey. No compensation was offered to any participant. In total, 1500 students, in 95 sample classes across 30 sample schools, participated in the study.

### 2.2. Outcome Measures

The mental health of students was measured by a self-reported Chinese-language version of the 25-item Strengths and Difficulties Questionnaire (SDQ), which has been widely used in low-, middle-, and high-income settings around the world, for assessing children and adolescents’ mental health problems [50,51,52]. It has been adapted and validated for use in China, demonstrating strong internal consistency (Cronbach’s alpha coefficient = 0.81) and high levels of validity (Pearson’s correlation coefficient = 0.71) [53,54]. The 25-item scale measures mental health according to two overall dimensions, including total difficulties and prosocial skills [55,56]. The total difficulties scale consists of four subscales (emotional problems, peer problems, behavioral problems, and hyperactivity), and each subscale comprises five items with three possible responses (“Not true”; “Somewhat true”; “Certainly true”). Fifteen of the total twenty subscale items are assigned values of 0, 1, 2, and the remaining five items are reversely scored as 2, 1, 0. Therefore, the total difficulties score ranges from 0 to 40, while each subscale of the difficulties scale has a possible total score of 0–10, with higher scores indicating more symptoms (and therefore indicating worse mental health). The prosocial skills scale also comprises five items with three possible responses for each item (“Not true” = 0; “Somewhat true” = 1; “Certainly true” = 2). The possible total score for this scale is also in the range of 0–10, with higher scores indicating better prosocial skills (and therefore indicating better mental health). 

In this paper, we primarily used the prosocial scale, as well as two amalgamated scales that are sometimes used instead of the four separate difficulties subscales described above. These alternative ten-item scales are called the ‘internalizing’ (comprising the emotional and peer problem subscales) and ‘externalizing’ subscales (comprising the conduct and hyperactivity subscales). Each of the amalgamated scores have possible ranges of 0–20, which can be added up to obtain the total difficulties scores. Using these two amalgamated scales may be preferable to using the four separate scales in community samples (as with the sample in the current study), whereas using the four separate scales may add more value in clinical samples [57].

Furthermore, dummy variables were calculated to classify scores on the subscales as either “abnormal” (1) or “normal” (0), according to the standardized cut-offs from the original 3-band solution [58]. We defined whether or not the internalizing and externalizing scores of the sample were in the normal range, using two different methods. In the first method, the dummy variable was defined as “abnormal” if both subscales on the internalizing (or externalizing) scales were in the abnormal range, according to the established cut-offs: that is, the internalizing (or the externalizing) scale score was defined as “abnormal” if the students obtained abnormal scores on both the emotional and conduct subscales (or on both the peer and hyperactivity subscales). Our second method was to define another dummy variable as having an abnormal score if either of the subscales on the internalizing (externalizing) scale were abnormal; that is, the internalizing (or the externalizing) scale score was defined as “abnormal” if the students obtained abnormal scores on either the emotional or conduct subscales (or on either the peer or hyperactivity subscales). Throughout the paper, we used the mean to fill in a small number of SDQ scores that lacked one or more subscale scores (<0.1% for sample students’ SDQ).

Student academic performance was measured by a 30-min standardized math test with 30 multiple-choice items. Each item on the test was assigned a score of 1 if the answer was correct, or 0 if it was incorrect, and the total test score was added up to 0–30. The test items for each sample grade were carefully designed, with assistance from educators working at the local education bureau, to ensure compliance with the national curriculum. The math testing scales had been used by the research team in several previous surveys to examine student academic performance in other parts of rural China [59,60]. We also pre-tested the exam multiple times to ensure its relevance in the sample schools and that time limits were appropriate. When we administered the exam in the sample schools, it was timed carefully and closely proctored by two trained enumerators. All test scores were then normalized according to the distribution of scores in each grade.

### 2.3. Family Functioning, Peer Victimization, and the Social Environment

The survey also collected data on a few other self-reported variables. Covariates in the study included basic demographic characteristics, including student gender (male or female) and age (which was divided into two quartiles according to the distribution of the sample: <11, ≥11). Information on whether students were boarding at school (“yes” or “no”), whether parents were alive (“yes” or “no”), and whether parents got divorced (“yes” or “no”) were also collected. To measure the education level of each student’s parents, we used high school attainment or above as a cut-off to create a dummy variable (>9 years education, “yes” or “no”). “Migrant father (mother)” (“yes” or “no”) indicated that the sample student’s father (mother) went out to work for more than 6 months in the past year, and those whose parents had both migrated were considered to be left-behind children (“yes” or “no”). To measure socioeconomic status, the questionnaire also asked whether or not the student’s household owned seven selected items included in the National Household Income and Expenditure Survey to create a family asset index, which we categorized into quartiles [61].

To measure the family functioning and cohesion of the sample students, we used the Family Functioning Assessment (FACES II-CV) questionnaire with 30 items, of which 14 were used to measure family cohesion, and 16 were used to estimate family flexibility [62]. Following the guidelines of the authors of the scale [63], the questionnaire covered both positive and negative aspects of family functioning, and each item was collected with a 5-point response, ranging from “almost never” (a score of 1) to “almost always” (a score of 5). The FACES II-CV questionnaire has been found to be reliable and valid for use among school-aged children and adolescents [64], and had been validated, previous to the current study, in mainland China [65]. In accordance with the ‘Circumplex Model’ of family systems, the families were further categorized as ‘extreme’, ‘mid-range’, and ‘balanced’ [65]. Dummy variables were used to define the three family categories for the overall scale [66,67].

The survey also collected information on several other characteristics of the social environment and behavior of the sample students. Information on bullying victimization was collected, using an international scale known as the Delaware Bullying Victimization Scale (Student Volume) (DBVS-S), consisting of four subscales named verbal, physical, social-relational, and cyberbullying [68]. A score for each subscale was derived by summing responses across the four items of each subscale. Each item was set with a 6-point Likert scale (1 = Never; 2 = Less than Once a Month; 3 = Once or Twice a Month; 4 = Once a Week; 5 = Several Times a Week; and 6 = Every Day). This scale has been used to describe the frequency of peer bullying victimization among children and adolescents [69], and it has been previously validated in China [70]. Following the method of prior studies [71], the responses were categorized into two groups, based on whether students had experienced bullying victimization at school or not. In addition to information about bullying, the respondents were asked to indicate how many minutes they spent in a typical school day engaging in screen time (i.e., recreational activities performed using a smartphone). The responses were then categorized into dummy variables, using 30 min (for smartphone use) as the cut-off. Information on how often the students typically participated in group-based activities organized by their school was also collected, and the responses were coded into a dummy variable indicating whether or not they often chose to participate.

### 2.4. Statistical Analysis 

The statistical analysis in the paper followed three steps. 

Firstly, in the descriptive analysis, the summary statistics of the sample were reported, as well as the means and standard deviations of students’ mental health from three dimensions (internalizing, externalizing, and prosocial skills) and academic performance in math.

Secondly, the correlates of student mental health and control variables were examined. An ordinary least square (OLS) linear regression model was used to conduct the multivariate analysis, so as to further identify the social–environmental factors associated with student mental health.

Thirdly, the association between student mental health and academic performance in math was measured while controlling for student and family characteristics with the hierarchical linear multiple regressions. A series of variables were considered as potential confounders in our multivariate analysis, including student age and gender, whether the student boarded at school, parental education level, parental migration status, and family asset status. All regressions were estimated with class-level fixed effects controlled, and all standard errors were clustered at the class level to account for the nested nature of the data.

All analyses were performed in Stata 16.1 (StataCorp LP, College Station, TX, USA). Any *p*-values below 0.1 were considered statistically significant.

## 3. Results

### 3.1. Summary Statistics

Table 1 shows the summary statistics of the sample. Slightly under half of the sample students (48%) were female, while the mean age was 11 years old, and 51% of the sample students were over 11 years old. Fourteen percent (14%) of the sample students boarded in a dormitory at school. About 8% of our sample students had parents who were divorced. All children had at least one parent still alive, but 1.7% and 1.1% of the students had one deceased parent (father or mother, respectively). On average, the fathers of the sample students were 41 years old, while their mothers were 38 years old. Less than a quarter of parents had received more than nine years of education, including 23% of fathers and 14% of mothers. One fifth of the sample students (21%) were left-behind children, whose parents were both migrants and did not live at home. Specifically, migrants accounted for more than half of the fathers (55%) and about a quarter of the mothers (27%).

The analysis also produced point estimates of the key covariates of the study. Specifically, 18% of students were living in families in the ‘balanced’ category, while almost 30% were living in families in the ‘extreme’ category. More than half of the students (52%) lived in families in the ‘mid-range’ category. Nearly 85% of the sample reported being bullied at school, including verbal (77%), physical (69%), social–relational (65%), and cyberbullying (29%). The average phone-usage time of the sample students was 10.3 min per day, and 13.5% reported an average daily screen time exceeding half an hour. Finally, 68% of the sample indicated that they frequently participated in group activities organized by their school.

Additionally, the mean SDQ scores of the sample were reported and were classified according to the categories of the original 3-band solution (Table 2). The mean SDQ total difficulties score of the sampled students was 12.39 (SD = 5.35), obtained by adding emotional (mean = 3.53; SD = 2.36), conduct (mean = 2.45; SD = 1.73), hyperactivity (mean = 3.32; SD = 2.00) and peer (mean = 3.09; SD = 1.64) problem scores. Among the students in our sample, about 10% of the sample students had abnormal total difficulties scores, which was close to the norm indicated in the official SDQ scoring guide [58], but higher than that in other developed countries, such as 2.9% in Germany and 4.2% in the Netherlands [72]. In regard to specific subscales, the prevalence of abnormal scores was 11.0%, 9.2%, 12.2%, and 6.1% on the emotional problem, peer problem, conduct problem, and hyperactivity/inattention problem subscales, respectively. Moreover, 7.4% of the sample students had abnormal prosocial scores, with an average score of 7.66 (SD = 1.96).

### 3.2. OLS Regression of Factors Correlated with Student Mental Health

Our study showed the associations between social–environmental characteristics and student mental health (Table 3). We found that the SDQ internalizing and prosocial scores of girls were significantly higher than those of boys, with a difference of 0.58 points (*p* < 0.001—column 2) and 0.41 points (*p* < 0.001—column 3), respectively. Students who had a migrant father had a 0.30-point-lower prosocial score (*p* = 0.006) and a 0.36-point-higher externalizing score (*p* = 0.025) than their peers. Students with divorced parents and lower asset index scores were found to have higher levels of SDQ internalizing scores (0.77 points, *p* = 0.016, and 0.40 points, *p* = 0.034, respectively); that is, their internalizing problems were more serious.

Different measures of bullying were also associated with higher student SDQ internalizing scores, including verbal bullying (0.61 points, *p* = 0.009), physical bullying (0.75 points, *p* = 0.001), social-relational bullying (0.67 points, *p* = 0.002), and cyberbullying (1.14 points, *p* < 0.001) in column 2. Meanwhile, higher SDQ externalizing scores were related to verbal (0.91 points, *p* < 0.001), physical (0.69 points, *p* = 0.001) and cyberbullying (1.25 points, *p* < 0.001) in column 1. Experiencing social-relational bullying at school was associated with a 0.39-point decrease (*p* = 0.004—column 3) in prosocial scores, while cyberbullying was associated with a 0.26-point decrease (*p* = 0.041).

Additionally, phone time over 30 min per day was also positively associated with both externalizing (0.89 points, *p* < 0.001) and internalizing symptoms (0.60 points, *p* = 0.012). Students who often attended group activities, organized by the school, had lower externalizing scores (−0.53 points, *p* = 0.002) and higher prosocial scores (0.58 points, *p* < 0.001).

However, several social–environmental characteristics did not have any associations with the student SDQ sub-scale scores in our model (Table 3). These characteristics included student age, boarding status, parental age, parental education level, and family functioning.

### 3.3. Association between Student Mental Health and Academic Performance in Math

Our main finding was that student mental health as measured by the SDQ subscales was significantly associated with academic achievement in math. We first conducted a hierarchical regression analysis to determine whether student SDQ scores predicted their academic performance in math (Table 4). We found that a one-point increase in the student SDQ externalizing score was associated with a 0.05 SD decrease in the standardized math score (*p* < 0.001), while a one-point increase in the internalizing score was associated with a 0.03 SD decrease (*p* = 0.002). In addition, a one-point increase in the student SDQ prosocial score was associated with a 0.02 SD increase in the standardized math score (*p* = 0.084). Meanwhile, being cyberbullied also significantly predicted math test scores among rural students (−0.40 SD, *p* < 0.001). After controlling for demographics, family functioning, peer victimization, and other social–environmental factors, the R-squared increased by 3.5% when the three SDQ subscales score were added to the model.

Secondly, we found that students with externalizing problems (that is, students whose peer and hyperactivity subscale scores were both in the abnormal range) had significantly lower math scores (−0.46 SD, *p* = 0.011) (Table 5, column 4). Similarly, students with internalizing problems (that is, students whose emotional and conduct subscale scores were both in the abnormal range) also had significantly lower math scores (−0.43 SD, *p* = 0.009).

This result was still robust when using a second method of calculating abnormal externalizing and internalizing scores (that is, by assigning an abnormal score if either subscale score of the externalizing/internalizing scales was abnormal) (Table 5 column 6). Using this method, the coefficients of the abnormal externalizing, internalizing, and prosocial scores changed slightly to −0.35 SD (*p* < 0.001), −0.23 SD (*p* = 0.001), and −0.33 SD (*p* = 0.001), respectively, but the direction of the coefficients was robust.

There was a clear trend between SDQ internalizing, externalizing, prosocial scores, and mean math performance of the sample students across a range of values (Figure 1). Students who had higher scores in the internalizing (Figure 1a) and externalizing problem (Figure 1b) scales both tended to have lower math scores. On the contrary, students with higher prosocial behavior scores showed a clear trend of scoring higher on the math test (Figure 1c). An additional figure shows the trend of math scoring with SDQ subscales in more detail (see Figure 1).

## 4. Discussion

This paper was one of the first to explore the levels and correlates of mental health strengths and difficulties across three dimensions (externalizing symptoms, internalizing symptoms, and prosocial behavior), and their association with academic performance in math among primary and secondary school students in rural China. We first measured the level of student mental health using the internationally validated SDQ scale and found that the mental health of students in western rural China was worse than that reported in almost all other identified studies of similarly aged students both inside China [73] and outside China [74]. Next, we identified social–environmental factors that were associated with student mental health conditions in the adjusted equation, including paternal migration status, bullying victimization, phone-using time, activity participation, and other student characteristics. However, we did not find any associations between student SDQ subscale score and other factors, such as student age, boarding status, parental age, parental education level, and family functioning. Finally, the main result of our study was that there was a strong and positive association between all dimensions of student mental health measured by the SDQ (in terms of lower scores on the internalizing and externalizing scales, and higher scores on the prosocial scale) and academic performance in math, which persisted even when controlling for other factors.

Our study made several contributions to the existing literature. Firstly, as one of the first studies in rural China to rely on the multiple dimensions of a widely used, internationally valid scale for measuring mental health, it revealed that the SDQ scores of students in rural China may be among the worst globally. Most studies that have used the SDQ among similarly aged youth in developed countries—including the United States [74], the Netherlands [75], Germany [76], and Australia [77]—have reported lower total difficulties scores than in our sample, which indicates that our samples had worse behavioral difficulties. However, the mean prosocial score of our sample was similar to the outcomes of students in other countries, such as the United States (mean = 7.43) [78], the Netherlands (mean = 7.98) [75], and Australia (mean = 8.03) [79]. The mean SDQ total difficulties score in this study’s rural sample was also higher than that reported in one study of migrant and left-behind children in Wuhan and its surrounding rural areas in 2014, though the prosocial scores were demonstrably higher [73]. For future studies in China, it will be necessary to conduct research on samples from both urban and rural areas to reach a better understanding of urban–rural differences in student mental health across different dimensions.

Secondly, our study was one of the first in the developing countries to use an international scale (SDQ) to measure mental health across three dimensions, and its association with academic performance in math, while controlling for other sociodemographic factors. The results of our study provide evidence that mental health may play an important role in the academic success of students living in low resource, developing contexts, thus contributing to the existing literature by demonstrating the association of academic performance in math with different dimensions of mental health (as opposed to just the symptoms of one specific disorder). Both internalizing and externalizing symptoms were negatively associated with arithmetic and mathematics. Specifically, students with internalizing problems (including emotional and peer problems) tended to have poorer math test scores, and the association was stronger when they had abnormal emotional and peer problem scores at the same time. Similarly, students with either conduct or hyperactivity external problems also displayed poorer academic performance in math, and those with abnormal scores on both subscales performed even worse. In addition, our study showed that prosocial skills were positively associated with academic performance in math. Although mental health and academic performance may have a bi-directional relationship [80], these results indicate that multiple aspects of mental health may be linked to academic performance. Our study thus suggests that it is important for interventions to not only address symptoms of internalizing disorders like depression (which is often the target of interventions), but also to aim for reducing externalizing symptoms and improving prosocial outcomes.

Another way in which our study contributes to the existing literature is that it highlights the social–environmental factors that are associated with student mental health in the context of rural China. Our results indicate that paternal migration is associated with significantly higher externalizing difficulties scores, which is in line with earlier findings [81,82]. Specifically, past studies have generally shown that both current and previous parental absence may decrease the care, stimulation, and communication that left-behind children receive from or have with their parents, thereby leading to the emergence of psychological and behavioral problems [83,84,85,86,87]. Furthermore, past research has shown that the mental health status of left-behind children under a grandparent’s guardianship is generally worse than that under a single-parent guardianship [88]. One surprising finding was that maternal migration status had no significant association with student mental health, which is in contrast to past findings that, when compared to paternal migration, maternal migration generally had a worse impact on child mental health [89,90]. It has previously been hypothesized that maternal migration matters more because mothers tend to engage in more communication and interaction with their children; however, in recent years, the participation of fathers in such activities has also been increasing [90,91,92]. Although our findings have considered a number of social and familial background variables, which differ from those of most previous studies, there may be other factors influencing these outcomes, such as respondent age, survey time, methods of survey administration, and other social determinants of health. Further research is necessary to better understand the way that parenting styles in migrant families may be evolving—as it may be undesirable and unfeasible to prevent parents from engaging in work-related migration [81]—and how this may determine the differential impact of which parent is absent, which can inform the design of supportive interventions [22,81,93].

In addition to paternal migration, other significant correlates were social behaviors, such as phone usage of more than half an hour a day (which had a positive association with externalizing and internalizing scores—indicating more behavioral problems), and attending group activities (which had a negative association with externalizing scores, and a positive association with prosocial scores—indicating fewer behavior problems) (see Table 3). These findings aligned with the conclusions in past literature that the structured and unstructured activities pursued by adolescents during after-school hours are critical for their mental health and wellbeing [94,95]. Past research has shown that screen time may be linked to both adolescent externalizing and adolescent internalizing behavior problem symptoms [96,97], while participation in group-based, extracurricular activities is generally associated with a lower prevalence of behavioral problems [98]. Time spent on one activity may also displace time spent on another: for example, the time spent on digital devices takes time from other activities and health-related behaviors, such as physical activity, supportive social interactions, or staying-on task at work or school [99]. Additionally, the positive association we found between lower family income and internalizing symptoms (see Table 3) was consistent with previous studies which indicated that economic hardship has a direct effect on children’s internalizing symptoms by reducing quality of life and increasing a child’s sense of helplessness and shame [100], as well as aligning with research that has shown that poverty is one of the strongest predictors of mental health outcomes for children [101]. In the past literature, parental education was identified as another aspect that predicts the quality of a child’s social environment by influencing parenting style and the quality of parent–child interactions [102]; however, we did not find significant associations with parental educational level among our sample, which may be related to the fact that many parents in our sample migrated out for work and could not interact frequently with their children in person. 

Finally, we also explored the role of bullying as a social–environmental determinant of student mental health and academic performance, as more than 80% of the students in our sample had experienced bullying. Our study demonstrated that all forms of bullying victimization—verbal bullying, physical bullying, social-relational bullying, and cyberbullying—were broadly associated with poorer student mental health, including higher internalizing scores (for all four types of bullying), higher externalizing scores (for verbal, physical, and cyberbullying), and lower prosocial scores (for social-relational and cyberbullying). Of course, it is possible that the relationship between bullying victimization and mental health may be bi-directional [103], as students who have traits such as strong prosocial skills may be less likely to be a target of bullying, while frequent bullying may also harm student mental health due to increased exposure to stressors. The extent to which there are causal relationships between different forms of bullying and student mental health is worth exploring in future research, as incidences of bullying continue to increase among students in both developing and developed regions [104,105,106].

We note several limitations of this study. Firstly, as this study used a cross-sectional and non-experimental design, the interpretation of the results could not assume causal relations among social–environmental factors, student mental health, and academic performance in math. Mental health and academic performance may have a bidirectional relationship, as a student’s academic success may reduce externalizing or internalizing symptoms, or it may improve their prosocial skills. For this reason, future research using randomized control trials or prospective longitudinal studies to enable observation of trajectories could be conducted to determine the directional pathway between mental health and students’ academic achievement. Secondly, although the sample was large, the area where we collected the sample was in the western part of China, which has the lowest levels of economic development in China; therefore, the results obtained may not be generalizable to other populations or contexts. Finally, the mental health of parents and teachers—whose psychological conditions are often closely related to the development of children and adolescents [107,108,109]—was not considered in the current study, and this may be an important social–environmental consideration for future studies. 

## 5. Conclusions

Using a sample of 1500 students selected from 30 primary and lower secondary schools in a rural region of northwestern China, we measured student mental health in terms of externalizing, internalizing, and prosocial behaviors; we identified the social–environmental correlates of mental health; and we measured the association of mental health with academic performance in math. The results show that the sample students had poor mental health when compared to most past studies inside and outside China. Bullying, screen time, and paternal migration were negatively associated with students’ mental health, while involvement in school group-based activities and higher family income were positively associated. This study also showed a strong positive association between poor mental health and academic performance in math: specifically, abnormal externalizing (internalizing) scores were associated with a 0.34 SD (0.22 SD) lower math score (*p* < 0.01), and abnormal prosocial scores were associated with a 0.35 SD lower math score (*p* < 0.01).

By exploring behavioral strengths and difficulties and their association with academic performance in math among primary and secondary school students in rural China, our study added to the existing literature on the links between social environment, mental health, and academic performance in math among students in developing countries. The results show that the mental health problems of students in poor, rural areas in China need urgent attention, both to improve the psychological wellbeing of rural students and to improve their academic performance, which lags behind their peers in more developed, urban areas of China. Our findings provide insight into the role that interventions, designed to reduce mental health symptoms and raise prosocial skills, can play in improving the academic performance of students in underdeveloped areas. Future studies should consider the determinants and correlates of mental health across multiple dimensions, and over the long term, to obtain a better understanding of how student mental health can positively impact academic outcomes.

## Figures and Tables

**Figure 1 healthcare-10-01642-f001:**
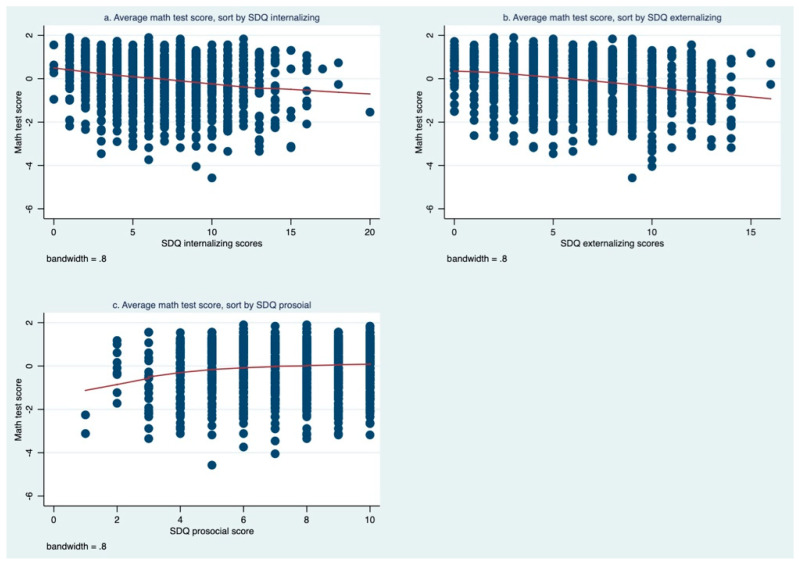
Mean of math test score by SDQ internalizing, externalizing, and prosocial scores: (**a**) average math test score sorted by SDQ internalizing; (**b**) average math test score sorted by SDQ externalizing; (**c**) average math test score sorted by SDQ prosocial. Throughout the figures, we deleted extreme values (which accounted for less than 1% of the sample), to better present the trend of changes in the SDQ subscale scores and math test scores.

**Table 1 healthcare-10-01642-t001:** Summary statistics of sample characteristics.

	Obs.	Mean	Std. Dev.	Min	Max
Gender (1 = female)	1500	47.8%	0.5	0	1
Age, years	1500	11.468	1.646	7.917	15.5
Over 11 years old (1 = Yes)	1500	51.3%	0.5	0	1
Boards at school (1 = Yes)	1500	14.3%	0.35	0	1
Parents are divorced (1 = Yes)	1500	7.7%	0.267	0	1
Father has passed away (1 = Yes)	1500	1.7%	0.131	0	1
Mother has passed away (1 = Yes)	1500	1.1%	0.103	0	1
Father’s age, years	1500	40.967	6.161	26	69
Mother’s age, years	1500	37.932	5.636	21	64
Father’s education level (>9-year, 1 = Yes)	1500	23.3%	0.423	0	1
Mother’s education level (>9-year, 1 = Yes)	1500	14.3%	0.35	0	1
Left-behind child (both out, 1 = Yes)	1500	20.7%	0.405	0	1
Migrant father (1 = yes)	1500	54.7%	0.498	0	1
Migrant mother (1 = yes)	1500	27.4%	0.446	0	1
Standardized values of math raw score	1500	−0.01	1.039	−4.573	1.902
Balanced family (1 = Yes)	1500	18.4%			
Mid-range family (1 = Yes)	1500	51.8%			
Extreme family (1 = Yes)	1500	29.8%			
Being bullied at school	1500	85.2%	0.355	0	1
Being verbally bullied (1 = Yes)	1500	76.6%	0.424	0	1
Being physically bullied (1 = Yes)	1500	69%	0.463	0	1
Being social-relationally bullied (1 = Yes)	1500	65.3%	0.476	0	1
Being cyberbullied (1 = Yes)	1500	28.7%	0.452	0	1
Phone time after school (mins)	1500	10.274	21.117	0	300
Daily phone time over 30 min (1 = Yes)	1500	13.5%	0.342	0	1
Often attend group activities (1 = Yes)	1500	68.4%	0.465	0	1

**Table 2 healthcare-10-01642-t002:** Summary statistics of SDQ.

	Obs.	Mean	Std. Dev.	Min	Max
SDQ total difficulties score	1500	12.389	5.354	1	34
SDQ internalizing scores	1500	6.619	3.232	0	20
SDQ emotional problems score	1500	3.533	2.362	0	10
SDQ peer problems score	1500	3.085	1.64	0	10
SDQ externalizing scores	1500	5.771	3.09	0	16
SDQ conduct problems score	1500	2.451	1.731	0	9
SDQ hyperactivity score	1500	3.319	2.003	0	10
SDQ prosocial score	1500	7.661	1.959	1	10
**Abnormal mental health status, according to SDQ cut-off points**					
Abnormal SDQ total difficulties (1 = Yes)	1500	10.3%	0.304	0	1
Abnormal SDQ emotional problems (1 = Yes)	1500	11.5%	0.319	0	1
Abnormal SDQ peer problems score (1 = Yes)	1500	9.2%	0.289	0	1
Abnormal SDQ conduct problems (1 = Yes)	1500	12.2%	0.327	0	1
Abnormal SDQ hyperactivity score (1 = Yes)	1500	6.1%	0.24	0	1
Abnormal SDQ prosocial score (1 = Yes)	1500	7.4%	0.262	0	1

**Table 3 healthcare-10-01642-t003:** The social–environmental factors associated with children’s mental health.

Variables	SDQ Externalizing Score	SDQ Internalizing Score	SDQ Prosocial Score
	(1)	(2)	(3)
Gender, 1 = female, 0 = male	−0.01	0.58 ***	0.41 ***
	(0.15)	(0.16)	(0.10)
Over 11 years old (1 = Yes)	−0.20	−0.13	−0.13
	(0.26)	(0.28)	(0.17)
Board at school (1 = Yes)	0.36	0.44	−0.05
	(0.29)	(0.30)	(0.19)
Father has passed away (1 = Yes)	0.17	0.54	−0.12
	(0.59)	(0.62)	(0.39)
Mother has passed away (1 = Yes)	0.16	0.06	0.56
	(0.74)	(0.78)	(0.49)
Parents are divorced (1 = yes)	0.33	0.77 **	−0.10
	(0.30)	(0.32)	(0.20)
Father’s age, years	0.04 *	0.02	−0.00
	(0.02)	(0.02)	(0.01)
Mother’s age, years	0.01	−0.00	−0.00
	(0.02)	(0.02)	(0.02)
Father’s education level (>9 years) (1 = Yes)	−0.12	−0.18	0.07
	(0.20)	(0.21)	(0.13)
Mother’s education level (>9 years) (1 = Yes)	−0.10	−0.31	−0.15
	(0.24)	(0.25)	(0.16)
Migrant father (1 = yes)	0.36 **	0.16	−0.30 ***
	(0.16)	(0.17)	(0.11)
Migrant mother (1 = yes)	−0.03	0.15	−0.12
	(0.19)	(0.20)	(0.13)
Midrange family (1 = Yes)	−0.22	−0.25	0.08
	(0.21)	(0.22)	(0.14)
Extreme family (1 = Yes)	−0.26	−0.20	0.24
	(0.23)	(0.24)	(0.15)
Being verbally bullied (1 = Yes)	0.91 ***	0.61 ***	−0.20
	(0.22)	(0.23)	(0.15)
Being physically bullied (1 = Yes)	0.69 ***	0.75 ***	0.02
	(0.21)	(0.22)	(0.14)
Being social-relationally bullied (1 = Yes)	0.24	0.67 ***	−0.39 ***
	(0.21)	(0.22)	(0.14)
Being cyberbullied (1 = Yes)	1.25 ***	1.14 ***	−0.26 **
	(0.19)	(0.20)	(0.13)
Daily phone time over 30 min (1 = Yes)	0.89 ***	0.60 **	−0.08
	(0.23)	(0.24)	(0.15)
Often attend group activities at school (1 = Yes)	−0.53 ***	−0.19	0.58 ***
	(0.17)	(0.18)	(0.11)
Family asset index (1 = bottom 25%)	−0.16	0.40 **	0.00
	(0.18)	(0.19)	(0.12)
Constant	2.25 ***	3.92 ***	8.04 ***
	(0.82)	(0.86)	(0.55)
Observations	1500	1500	1500
R-squared	0.230	0.227	0.155

Standard errors clustered at class level are in parentheses. *** *p* < 0.01, ** *p* < 0.05, * *p* < 0.1.

**Table 4 healthcare-10-01642-t004:** Hierarchical regression of SDQ score and academic performance in math.

Independent Variables	Standardized Values of Math Test Scores					
	(1)	(2)	(3)	(4)	(5)	(6)
**Step 1: Control variables**						
Gender, 1 = female, 0 = male	−0.13 **	−0.15 ***	−0.15 ***	−0.12 **	−0.17 ***	−0.15 ***
	(0.05)	(0.05)	(0.05)	(0.05)	(0.05)	(0.05)
Over 11 years old (1 = Yes)	−0.22 **	−0.20 **	−0.22 **	−0.21 **	−0.20 **	−0.22 **
	(0.09)	(0.09)	(0.09)	(0.09)	(0.09)	(0.09)
Left-behind child (1 = Yes) (both parents out)	0.01	0.02	0.04	0.03	0.03	0.05
	(0.07)	(0.07)	(0.07)	(0.07)	(0.07)	(0.07)
Boarding at school (1 = Yes)	−0.18 *	−0.15	−0.12	−0.13	−0.14	−0.11
	(0.10)	(0.10)	(0.10)	(0.10)	(0.10)	(0.09)
Daily phone time over 30 min (1 = Yes)	−0.18 **	−0.16 **	−0.11	−0.14 *	−0.16 **	−0.10
	(0.08)	(0.08)	(0.08)	(0.08)	(0.08)	(0.08)
Often attends group activities at school (1 = Yes)	0.24 ***	0.21 ***	0.18 ***	0.20 ***	0.19 ***	0.16 ***
	(0.06)	(0.06)	(0.06)	(0.06)	(0.06)	(0.06)
Father has passed away (1 = Yes)	0.37 *	0.35 *	0.37 *	0.38 *	0.36 *	0.38 **
	(0.20)	(0.20)	(0.19)	(0.20)	(0.20)	(0.19)
Mother has passed away (1 = Yes)	0.02	0.10	0.11	0.11	0.08	0.09
	(0.25)	(0.25)	(0.24)	(0.25)	(0.25)	(0.24)
Parents are divorced (1 = Yes)	−0.05	−0.04	−0.03	−0.01	−0.04	−0.01
	(0.10)	(0.10)	(0.10)	(0.10)	(0.10)	(0.10)
Father’s age, years	−0.01	−0.01	−0.01	−0.01	−0.01	−0.01
	(0.01)	(0.01)	(0.01)	(0.01)	(0.01)	(0.01)
Mother’s age, years	0.00	0.00	0.00	0.00	0.00	0.00
	(0.01)	(0.01)	(0.01)	(0.01)	(0.01)	(0.01)
Father’s education level (>9 years) (1 = Yes)	0.12 *	0.10	0.09	0.09	0.09	0.08
	(0.07)	(0.07)	(0.07)	(0.07)	(0.07)	(0.07)
Mother’s education level (>9 years) (1 = Yes)	0.17 **	0.16 *	0.15 *	0.14 *	0.16 **	0.14 *
	(0.08)	(0.08)	(0.08)	(0.08)	(0.08)	(0.08)
Family asset index, 1 = bottom 25%	−0.05	−0.05	−0.06	−0.03	−0.05	−0.05
	(0.06)	(0.06)	(0.06)	(0.06)	(0.06)	(0.06)
**Step 2: Bullying status**						
Being verbally bullied (1 = Yes)		0.09	0.15 **	0.12	0.10	0.16 **
		(0.07)	(0.07)	(0.07)	(0.07)	(0.07)
Being physically bullied (1 = Yes)		−0.10	−0.05	−0.06	−0.10	−0.04
		(0.07)	(0.07)	(0.07)	(0.07)	(0.07)
Being social-relationally bullied (1 = Yes)		−0.02	−0.01	0.01	−0.01	0.02
		(0.07)	(0.07)	(0.07)	(0.07)	(0.07)
Being cyberbullied (1 = Yes)		−0.40 ***	−0.32 ***	−0.35 ***	−0.39 ***	−0.30 ***
		(0.06)	(0.06)	(0.06)	(0.06)	(0.06)
**Step 3: SDQ subscales**						
SDQ externalizing scores			−0.07 ***			−0.05 ***
			(0.01)			(0.01)
SDQ internalizing scores				−0.05 ***		−0.03 ***
				(0.01)		(0.01)
SDQ prosocial score					0.04 ***	0.02 *
					(0.01)	(0.01)
Constant	0.02	0.24	0.40	0.42	−0.10	0.29
	(0.27)	(0.27)	(0.27)	(0.27)	(0.29)	(0.29)
Class fixed effect	Yes	Yes	Yes	Yes	Yes	Yes
Observations	1500	1500	1500	1500	1500	1500
R-squared	0.201	0.231	0.260	0.247	0.237	0.266
R-squared change		0.030	0.029	0.016	0.006	0.035
F-test		13.513 ***	53.945 ***	28.635 ***	10.562 ***	22.223 ***

Standard errors clustered at class level are in parentheses. *** *p* < 0.01, ** *p* < 0.05, * *p* < 0.1

**Table 5 healthcare-10-01642-t005:** OLS regression of SDQ score and academic performance in math.

Independent Variables	Standardized Values of Math Test Scores					
	(1)	(2)	(3)	(4)	(5)	(6)
SDQ externalizing scores	−0.06 ***	−0.05 ***				
	(0.01)	(0.01)				
SDQ internalizing scores	−0.04 ***	−0.03 ***				
	(0.01)	(0.01)				
Abnormal SDQ externalizing score (Both) (1 = Yes)			−0.67 ***	−0.46 **		
			(0.18)	(0.18)		
Abnormal SDQ internalizing score (Both) (1 = Yes)			−0.52 **	−0.43 **		
			(0.17)	(0.16)		
Abnormal SDQ externalizing score (Any) (1 = Yes)					−0.45 ***	−0.35 ***
					(0.07)	(0.07)
Abnormal SDQ internalizing score (Any) (1 = Yes)					−0.31 ***	−0.23 ***
					(0.07)	(0.07)
SDQ prosocial score	0.03 **	0.02 *				
	(0.01)	(0.01)				
Abnormal SDQ prosocial score (1 = Yes)			−0.44 ***	−0.36 ***	−0.39 ***	−0.33 ***
			(0.10)	(0.10)	(0.10)	(0.10)
Bulling status controlled	Yes	Yes	Yes	Yes	Yes	Yes
Fixed effect in class level	Yes	Yes	Yes	Yes	Yes	Yes
Control variables	No	Yes	No	Yes	No	Yes
Constant	0.42 **	0.29	0.17	0.31	0.23	0.25
	(0.21)	(0.29)	(0.17)	(0.27)	(0.16)	(0.27)
Observations	1500	1500	1500	1500	1500	1500
R-squared	0.227	0.266	0.190	0.247	0.218	0.262

Standard errors clustered at class level are in parentheses. *** *p* < 0.01, ** *p* < 0.05, * *p* < 0.1

## Data Availability

The data presented in this study are available on request from the corresponding author. The data are not publicly available, as they contain sensitive information on the mental health of children and adolescents in rural China.

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
