# Peer review of "Behavioral Strengths and Difficulties and Their Associations with Academic Performance in Math among Rural Youth in China"

_healthcare, 2022, doi:10.3390/healthcare10091642_

Round 1

Reviewer 1 Report

The manuscript entitled Behavioral strengths and difficulties and their associations with academic performance among rural youth in China (healthcare-1854433) describes exploratory findings between correlates of mental health on academic functioning using the Strengths and Difficulties Questionnaire (SDQ) among rural students in China. This study specifically examines the relationship between various domains of mental health specifically internalizing, externalizing, and prosocial behaviors) and academic functioning. Overall, this article provides a comprehensive review of imperative considerations. Strengths of the study include that the introduction sufficiently reviews current literature, the robust sample size, and the comprehensive steps taken to develop the method of evaluating mathematical academic functioning. However, there are substantial recommendations for the Methods and Discussion sections, which will further bolster the strength of the proposed article. Thus, this reviewer offers the following recommendations before publication.

Materials and Methods

1.     Please include details explaining how the authors obtained data from surveys. More specifically, did the students complete the surveys themselves or was data collected via parent-report?

2.     Please include the compensation participants received for engaging in this study.

3.     Line 144, typo. The word currently reads “item2s”.

4.     Regarding the SDQ, please provide reliability and validity indices for the scale and subscales, if available.  

5.     The authors detail their process for developmental their assessment of math skills on Lines 177-187.  Please also provide additional details specific to the number of items and scoring process.

 Results:

1.     The rate in which students endorse an “Abnormal” score across the SDQ subscales is low, falling at 12.2% or less. The authors note that the SDQ scores are “much higher” (Lines 276) relative to corresponding scores in other developed countries, however it is recommended that the authors calculate the statistical significance to further support their statement.

2.     The results of the current study highlights intriguing and relevant findings, however the statistical methods could be advanced to demonstrate more meaningful interpretations, extending beyond reporting correlations and associations. Thus, the reviewer strongly recommends that the authors consider analyzing their data using hierarchical linear multiple regressions to further evaluate predictive strengths and to generate mediation and moderation models. This method would be particularly beneficial allowing the authors to identify the differential predictive strength of each of the subscales in this study.

a.     Furthermore, considering that 85.2% of the sample report being bullied, it is recommended that the authors include this variable in their step-wise analyses during the multiple linear regression to further assess the variable’s predictive strength in impacting academic functioning.

Discussion:

1.     On lines 406-411, the authors offer a hypothesis discussing their finding contrary to previously established studies. In addition to the hypothesis discussed, it is recommended that the authors include and address the impact of being a Left Behind Child on mental health. It is also recommended that the authors include literature to explain how Left Behind Children are customarily cared for.

2.     The authors discuss the relationship between externalizing and internalizing scores and other demographic information in lines 415 – 423. This paragraph could also be further developed, as it is severely lacking in its current state. More specifically, it is recommended that the authors further parse apart and explain previous literature specific to 1) explain the role in the use of phone use, 2) role of engaging in extracurricular activities, and 3) the relationship between poverty and parental education level.

Reviewer 2 Report

This study examines the influences of behavioral and mental factors on academic performance in mathematics and provides valuable knowledge. However, the following points should be corrected.

1. The expression "impact" on line 19 is misleading to the reader. Since this is a cross-sectional study, the expression should be revised so that it does not suggest causality.

2. The details of how the 30 schools targeted in "2.1 Sampling" were selected from the sample list should be described. At least a description of whether it was a random sample and why 30 schools is required.

3. The rationale and explanation should be added as to whether "a 30-minute standardized math test" on line 176 is a comparable index for each grade.

4. The document on lines 249-251 is gibberish.

5. Were all the parents of the target children still alive? Wasn't there a single parent due to divorce or bereavement? Since these conditions affect the quality and validity of the study, please add them accurately.

6. Does "most past studies inside and outside of China" on line 454 refer to the information listed in Table S1?

7. In this study, since academic performance is evaluated only in arithmetic and mathematics, it should be specified as "academic performance in math" in the title and the entire text.

Reviewer 3 Report

The abstract does not clearly and concisely support all the concepts that are investigated in the study. I recommend that the abstract be revised appropriately for the study presented.

Furthermore, the conclusions should be broadened. The scope of future prospects must be emphasized.

Round 2

Reviewer 2 Report

I saw the manuscript this time and felt that it was improved appropriately. I think that it is good to judge that it can be published.